# Therapeutic Potential of Deflamin against Colorectal Cancer Development and Progression

**DOI:** 10.3390/cancers14246182

**Published:** 2022-12-14

**Authors:** Sara Silva, Ana Cavaco, Bianca Basso, Joana Mota, Raquel Cruz-Duarte, Miguel Costa, Lara Carvalho, Ana Lima, Luis Costa, Ricardo Ferreira, Marta Martins

**Affiliations:** 1Instituto de Medicina Molecular—João Lobo Antunes, Faculdade de Medicina, Universidade de Lisboa, 1649-028 Lisbon, Portugal; 2Faculty of Veterinary Medicine, Lusófona University, 1749-024 Lisbon, Portugal; 3LEAF—Linking Landscape, Environment, Agriculture and Food, Instituto Superior de Agronomia, Universidade de Lisboa, Tapada da Ajuda, 1349-017 Lisbon, Portugal; 4Oncology Division, Hospital de Santa Maria, Centro Hospitalar Lisboa Norte, 1649-028 Lisbon, Portugal

**Keywords:** deflamin, colorectal cancer, MMP-2, MMP-9

## Abstract

**Simple Summary:**

We have previously identified deflamin, an oligomeric protein isolated from the white lupine seeds (*Lupinus albus*) with anti-MMPs and anti-inflammatory properties. Given the involvement of MMPs and inflammation in the carcinogenesis process, we aimed to assess deflamin’s role in cancer development and progression. Using colorectal cancer cell lines and zebrafish xenotransplant models, we found that deflamin exhibits anti-MMP-2 and anti-MMP-9 activity, being able to reduce tumor size and metastasis formation in vivo. Deflamin was shown to impair cancer cell migration and invasion, as well as collagen remodeling and angiogenesis in the tumor microenvironment, highly impacting cancer behavior. Overall, our results unravel the nutraceutical potential of deflamin in colorectal cancer treatment.

**Abstract:**

Matrix metalloproteinases (MMPs) are proteolytic enzymes that play a crucial role in tumor microenvironment remodeling, contributing to inflammatory and angiogenic processes, and ultimately promoting tumor maintenance and progression. Several studies on bioactive polypeptides isolated from legumes have shown anti-migratory, anti-MMPs, and anti-tumor effects, potentially constituting novel strategies for both the prevention and progression of cancer. In this work, we investigated the anti-tumor role of deflamin, a protein oligomer isolated from white lupine seeds (*Lupinus albus*) reported to inhibit MMP-9 and cell migration in colorectal cancer (CRC) cell lines. We found that deflamin exerts an inhibitory effect on tumor growth and metastasis formation, contributing to increased tumor apoptosis in the xenotransplanted zebrafish larvae model. Furthermore, deflamin resulted not only in a significant reduction in MMP-2 and MMP-9 activity but also in impaired cancer cell migration and invasion in vitro. Using the xenograft zebrafish model, we observed that deflamin inhibits collagen degradation and angiogenesis in the tumor microenvironment in vivo. Overall, our work reveals the potential of deflamin as an agent against CRC development and progression.

## 1. Introduction

Colorectal cancer (CRC) is the third most common and the second leading cause of death by cancer worldwide [1]. As CRC patients die of metastatic disease, prevention of the development of metastasis is essential to improve cure rates.

To metastasize, a tumor cell has to invade the surrounding tissue, enter the bloodstream, survive in circulation, and extravasate and colonize the distant organ, a process that requires multiple interactions between the malignant cell and its microenvironment [2]. Matrix metalloproteinases (MMPs) comprise a large family of homologous zinc-dependent endoproteases, several of which play critical roles in this process, not only by degrading the extracellular matrix (ECM), allowing tumor dissemination and seeding but also by promoting angiogenesis and inflammation [3]. Therefore, increased MMP expression was detected in malignant tissues and was correlated with metastatic spread and unfavorable prognosis in multiple types of cancer [4,5]. Expectedly, MMPs were seen as ideal pharmacological targets for cancer therapy with preclinical studies of synthetic MMP inhibitors (MMPIs) holding great promise [6,7]. However, clinical trials developed during the late 1990s and early 2000s were unsuccessful in showing MMPIs effect in reducing tumor burden or improving overall survival, in addition to demonstrating severe side effects [8,9,10]. Further studies led to the conclusion that some MMPs, such as MMP-8, have anti-tumor effects, therefore the broad-spectrum of MMPIs was shown to be counter-productive, ultimately resulting in tumor progression and overall intense toxicity [11]. Thereby, it has now become apparent that to successfully target MMPs in the setting of cancer therapy, in situ inhibition of specific MMPs is essential. In particular, inhibition of the gelatinases MMP-2 and MMP-9 is envisaged, given their involvement in the degradation of the ECM and basement membrane, but also due to their role in the proteolysis of cell adhesion molecules and other bioactive proteins. In this context, old drugs and common natural products are now being explored for their potential to inhibit MMP-2 and MMP-9. Importantly, compounds derived from foods have been showing encouraging results in this regard, for example, curcumin, a component of the South Asian spice turmeric, has been shown to decrease MMP-2, -9, and -14 expression in various cancers, leading to decreased MMP activity and decreased cancer cell migration and invasion [12,13,14]. Similarly, antioxidant polyphenols in common foods, such as trans-resveratrol and quercetin from grapes and wine, as well as oleuropein and hydroxytyrosol from olive oil have also been shown to decrease the expression and activity of MMP-9 and decrease cancer cell migration, invasion, and angiogenesis [15,16,17]. Experimental studies of animal models fed with legume seeds have shown to reduce both the incidence and the number of colon tumors by 50% [18] and clinical studies have now started to provide evidence that legume consumption can decrease the risk of CRC [19]. In agreement with these reports, the World Cancer Research Fund/American Institute for Cancer Research recognized the potential of legume consumption in CRC prevention, supporting the need for additional research in this area.

We have recently discovered deflamin (patent WO/2018/060528), an oligomeric polypeptide isolated from the edible seeds of white lupin (*Lupinus albus*), that reduces MMP-2 and MMP-9 activity in CRC cell lines, in a dose-dependent manner and with an IC50 of 10 µg/mL. Being of food origin, and an oligomer of two storage proteins in legume seeds, deflamin isolated from white lupine was found to be safe for consumption, without impairing gene expression nor exerting cytotoxicity [20,21,22,23]. Its gelatinase inhibitory features suggested that it can be used in pathologies related to enhanced gastrointestinal MMP-9 activity, namely cancer, and inflammatory diseases. So far, deflamin was only successfully tested as a nutraceutical in inflammatory bowel disease models. In mice models of colitis, it significantly inhibited colonic MMP-9 activities, whilst reducing inflammation and colitis-induced lesions, when administered orally as a lupin seed extract or as a food additive to wheat cookies [21,24]. These works showed that deflamin is not only an efficient MMP-9 and MMP-2 inhibitor, but it also is highly resistant to digestion and not absorbed by the digestive tract (our unpublished results). These features infer that deflamin holds the potential to act locally in the intestinal path, bypassing the problem of systemic toxicity associated with common MMPIs [20,21,22,23,24]. Moreover, since it is water-soluble and rather effortlessly isolated [20], deflamin can easily become a novel nutraceutical for pathologies related to aberrant MMP-9 activities, particularly in the digestive system, such as CRC [22]. However, despite its potential against this disease, deflamin has never been tested in more realistic cancer models. In this context, we sought to explore the nutraceutical anti-tumor potential of this polypeptide in CRC progression. Not only this is the first report of deflamin in CRC models, but it also opens a door to novel approaches to tackle this disease.

## 2. Materials and Methods

### 2.1. Cell Lines

The following colorectal carcinoma cell lines were used in this work: HT-29 (ECACC, no. 91072201), HCT116 (ATCC^®^ CCL-247), and SW480 (ATCC^®^ CCL-228). All cell lines were cultivated in DMEM media supplemented with 10% (*v*/*v*) FBS and 1% penicillin/streptomycin at 37 °C and 5% CO_2_, in a humidified atmosphere. Cells were maintained at a low passage and routinely tested for mycoplasma contamination by qPCR. Cell lines were validated by short tandem repeat (STR) profile.

### 2.2. Lentiviral Infection

For transduction of HCT116, SW480 and HT29 with tdTomato fluorescent protein, cells were infected with lentiviral particles containing FUdtTW plasmid. 2 × 10^5^ cells were seeded in 6-well plates and incubated overnight at 37 °C. The next day, a mixture containing complete growth medium with 5 µg/mL polybrene, and lentiviral particles was added to the cells. The medium was replaced 24 h after infection and cells were expanded and then sorted in BD FACSAria III with a 98% purity.

### 2.3. Deflamin Purification

Deflamin was isolated from mature and dried white lupin seeds (*Lupinus albus*) as described previously [20] Briefly, lupin seeds were milled to a fine powder and extracted using 50 mM of Tris-HCl buffer, pH 7.5 (1:10, *w*/*v*). The homogenate was centrifugated at 12,000× *g* for 30 min at 4 °C. The supernatant was collected, boiled for 10 min, and centrifugated at 12,000× *g* for 20 min at 4 °C. The supernatant was then made to pH 4.0 and centrifugated at 12,000× *g* for 20 min at 4 °C. The resulting pellet was resuspended in 40% (*v*/*v*) ethanol containing 0.4 M NaCl, and centrifugated at 13,500× *g*, 30 min, 4 °C. The supernatant was made to 90% (*v*/*v*) ethanol and left overnight at −20 °C. The following day, the mixture was centrifugated at 13,500× *g* for 30 min at 4 °C and the pellet, containing isolated deflamin, was resuspended in the smallest possible volume of milli-Q water. Desalting was performed with PD-10 Columns (GE Healthcare Life Science, Uppsala, Sweden) according to manufacturer recommendations and the final solution was collected, frozen at −20 °C, and lyophilized. The obtained deflamin reached 98% purity and the integrity of the protein was detected by polyacrylamide gel electrophoresis and subsequent Coomassie brilliant blue staining [20]. Deflamin was diluted in PBS for in vitro and in vivo experiments.

### 2.4. Cellular Viability Assay

CRC cells were seeded at a density of 1–2 × 10^4^ cells in 96-well plates. A total of 24 h after seeding, cells were treated with 0, 25, 50, or 75 µg/mL of deflamin. Every day, for 5 consecutive days, 1:10 of AlamarBlue reagent (Invitrogen, Waltham, MA, USA) was added to each well, and fluorescence was measured 2 h after (excitation 560 nm; emission 590 nm) in Infinite M200 Plate Reader (Tecan).

### 2.5. Cellular Apoptosis Assay

For apoptosis analysis, cells were seeded at a density of 1–2 × 10^4^ cells in 96-well plates. 24 h after seeding, cells were treated with 0, 25, 50, or 75 µg/mL of deflamin for 48 h. The measurement of caspase 3/7 activity was performed using the Apo-ONE^®^ Homogeneous Caspase-3/7 Assay kit (G7790, Promega, Madison, WI, USA) following manufacture instructions.

### 2.6. Zymography

In order to determine the anti-MMP role of deflamin in cancer cells, a zymographic analysis was performed as previously described [22]. Briefly, 12.5% polyacrylamide-SDS gels (*v*/*v*) were co-polymerized with 1% gelatin (*w*/*v*). CRC cell lysates (without and with deflamin treatment at 20 μg/mL, 40 μg/mL, 80 μg/mL) were treated with a nonreducing buffer containing 62.6-mM Tris–HCl pH 6.8, 2% (*w*/*v*) SDS, 10% (*v*/*v*) glycerol and 0.01% (*w*/*v*) bromophenol blue were loaded into each well of the SDS-gel. Electrophoresis was carried out vertically at 100 V and 20 mA per gel. Subsequently, gels were washed three times in 2.5% (*v*/*v*) Triton X-100 for 90 min each, to remove the SDS and incubated with a solution of 50-mM Tris–HCl pH 7.4, 5-mM CaCl_2_, 1-μM ZnCl_2,_ and 0.01% *w*/*v* sodium azide, for 48 h at 37 °C. After incubation, gels were stained with Coomassie Brilliant Blue G-250 0.5% (*w*/*v*) in 50% (*v*/*v*) methanol and 10% (*v*/*v*) acetic acid, for 30 min, and destained with a solution of 50% (*v*/*v*) methanol, 10% (*v*/*v*) acetic acid. The gelatin degradation bands (white bands against a blue background), denoting relative MMP gelatinolytic activity, were analyzed according to their intensity with UN-SCAN-IT gelTM 6.1 software (Silk Scientific Corporation, Orem, UT, USA) and the relative values expressed in % of untreated lysate control.

### 2.7. Wound Healing Migration Assay

The effect of deflamin on cancer cell migration was analyzed by the wound healing assay. CRC cell lines were seeded in a 24-well plate at a density of 1 × 10^5^ cells/well and cultured until a monolayer of ~85% confluence was reached. A central scratch-like gap was created with a pipette tip and a medium containing increasing concentrations of deflamin (0, 20, 40, or 80 μg/mL) was added to each well. The migration ability of the cancer cells was assessed by the capacity to close the “wound” after 48 h of deflamin treatment. The cell-free area in the well was calculated with the Fiji/ImageJ software, and the relative values were calculated in percentage of control condition area values.

### 2.8. 3D Cell Invasion Assay

The 3D invasion assay was performed as previously described [25]. Briefly, for the generation of spheroids, we combined 1/4 of the final volume of methylcellulose (6 mg/mL), 10% (*w*/*v*) FBS, 20 to 40 µg/mL collagen, 1 µg/mL mitomycin in DMEM medium. Five hundred cancer cells were then suspended in 50 µL of this spheroid formation medium and plated into a non-adherent 384-well plate. Cells were incubated for 24 h to allow the formation of an individual spheroid per well. The medium was then replaced by 50 µL of collagen matrix composed of two parts of collagen (3 mg/mL), 15% (*w*/*v*) FBS, 1 µg/mL mitomycin, 2% (*w*/*v*) NaOH (1 M) and DMEM. Deflamin, at a concentration of 50 µg/mL and 100 µg/mL, was added to the treated cells. After 1 h incubation, imaging of the spheroids was performed by fluorescence microscopy in a Zeiss LSM710 confocal microscope. This time point was accounted as 0 h. The next images were taken at the time points of 24, 48, and 72 h. Images were analyzed using the FIJI/ImageJ software. Invasive cells were counted between the inner perimeter (spheroid external border at 0 h) and the outer perimeter (the furthest invasive cell at each time point).

### 2.9. Cell Labeling

CRC cell lines at 70% confluence were stained with lipophilic dye Vybrant™ DiI cell labeling solution (V22885 from Invitrogen) at 4 µL/mL in PBS 1X for 10 min at 37 °C.

### 2.10. In Vivo Zebrafish Xenograft Model

Wild-type and transgenic Tg(kdrl:EGFP) zebrafish (*Danio rerio*) [26] embryos were provided by the zebrafish facility of Instituto de Medicina Molecular (iMM). For husbandry, adult zebrafish were maintained at 28.5 °C in a 10/14 h dark-light cycle, according to standard protocols of the European Animal Welfare Legislation, Directive 2010/63/EU (European Commission, 2016), following the Federation of European Laboratory Animal Science Associations (FELASA) guidelines and recommendations. All procedures in this study were performed in early life forms of zebrafish development, with embryos up to 120 h post-fertilization (hpf), that do not yet show the ability to feed themselves and are, therefore, considered unprotected under the European Animal Welfare Legislation, Directive 2010/63/EU (European Commission, 2016).

For zebrafish injection, HCT116 cells expressing dTomato fluorescent protein or labeled with DiI were used as indicated. 2.5 × 10^5^ cells/mL (approximately 800 cells per injection) were microinjected into the perivitelline space (PVS) of 48 hpf larvae previously anesthetized with tricaine 1.5% (*w*/*v*; Pharmaq), as described before [27]. Microinjections were performed under a stereo microscope (Leica S8 APO) using borosilicate glass microcapillaries attached to a microinjector (World Precision Instruments, Pneumatic Pico pump PV820) coupled with a micromanipulator (Narishige MN-153). At 24 h post-injection, successfully injected xenografts were treated with 100 μg/mL deflamin in E3 medium or E3 with PBS (controls) and incubated at 34 °C. The medium was replaced daily for three days. After this period, animals were fixed in 4% (*w*/*v*) paraformaldehyde overnight and stored at −20 °C in 100% (*v*/*v*) methanol.

### 2.11. Immunofluorescence

Frozen xenografts were re-hydrated in methanol series (75% > 50% > 25%) and then permeabilized in acetone at −20 °C. Xenografts were then washed in a buffer containing 1x PBS, 0.5% (*v*/*v*) Tween 20, 0.5% TritonX-100, and 100 mM glycine for 1 h at room temperature (RT), followed by blocking in 1x PBS, 1% *w*/*v* BSA, 1% *v*/*v* DMSO, 1% *v*/*v* TritonX-100 and 1.5% *w*/*v* FBS, for 1 h at RT. Next, xenografts were incubated with primary antibodies: rabbit monoclonal anti-cleaved caspase-3 (Asp175) (1:100, #9661 from Cell Signaling); and mouse monoclonal anti-GFP (1:100, #11814460001 from Roche) for 1 h at RT and overnight at 4 °C. The following day, secondary antibody incubation with goat Anti-rabbit IgG H&L Alexa Fluor^®^ 488 (1:400, #A-11008 from Invitrogen) and 50 μg/mL DAPI was performed for 1 h at RT and then overnight at 4 °C. Xenografts were mounted with Vectashield^®^ mounting media between two coverslips and stored at 4 °C for subsequent analysis.

Xenografts were imaged in a confocal microscope Zeiss LSM 710 with a Z-stack interval of 5 μm. Images were analyzed in FIJI/ImageJ software using the plugin cell counter. For tumor size calculation, three representative slices of the tumor from the top (Zfirst), the middle (Zmidle), and the bottom (Zlast) were analyzed and the number of cells calculated as the sum of cells in Zfirst, Zmidle, Zlast/total number of Z stacks × 1.5. The 1.5 correction number was estimated for these CRC cells that have nuclei with an average of 10–12 μm of diameter. The number of mitotic figures and activated caspase-3 were counted in every slice of the tumor (from Zfirst to Zlast) and the percentage was obtained by dividing the value by the total number of cells in tumor size.

Metastasis was counted in the caudal hematopoietic tissue and in the gills of zebrafish larvae.

For the analysis of angiogenesis, all images were obtained with a 7 μm interval Z-stack and two parameters were measured: vessel density (VD) and vessel infiltration (VI). Vessel density and vessel infiltration were assessed through Z-projections of corresponding images using the ImageJ Z-Projection tool and the area of eGFP fluorescent signal per tumor was quantified. To analyze the vessel infiltration, the superficial slices of the images were not considered (about 20% of total stacks).
$$\mathrm{VD}\;=\frac{\mathrm{eGFP}\;\mathrm{area}}{\mathrm{Tumour}\;\mathrm{area}}\times 100,\;\mathrm{VI}=\frac{\mathrm{eGFP}\;\mathrm{area}\;\mathrm{in}\;\mathrm{tumour}\;\mathrm{core}}{\mathrm{core}\;\mathrm{of}\;\mathrm{the}\;\mathrm{tumour}\;\mathrm{area}}\times 100$$

For degraded collagen analysis, the 5-FAM fluorescence area corresponding to collagen degradation points per tumor was measured, using Z-projections of corresponding images with ImageJ Z-projection tool. The value was normalized by the total tumor area.
$$\mathrm{Degraded}\;\mathrm{collagen}=\frac{\mathrm{eGFP}\;\mathrm{area}}{\mathrm{Tumour}\;\mathrm{area}}\times 100$$

### 2.12. Statistical Analysis

GraphPad Prism 8 software (Dotmatics, San Diego, USA) was used for the statistical analysis of in vivo experiments. Pared, unpaired t-test and one-way ANOVA were used to analyze in vitro and in vivo data, as indicated in figure legends. Results are presented as average ± standard error of the mean (SEM). The level of statistical significance was set as non-significant (NS); *, *p* < 0.05; **, *p* < 0.01; ***, *p* < 0.001; and ****, *p* < 0.0001. For all the statistical analysis *p* value as a confidence interval of 95%.

## 3. Results

### 3.1. Deflamin Exhibits Anti-Tumor Activity in Zebrafish CRC Xenotransplanted Tumors

In previous studies, we found that deflamin had anti-MMP-2 and -MMP-9 activities in CRC cell line HT-29, as well as anti-inflammatory role in mouse models of induced-intestinal colitis [21]. Therefore, we sought to investigate deflamin’s role in cancer development using the zebrafish larvae CRC xenograft model. This is a well-established tumorigenesis model that displays several advantages: it is a quick assay with cellular resolution and allows the evaluation of crucial hallmarks of cancer, such as tumor proliferation, and metastatic and angiogenic potentials [27]. Therefore, CRC cell line HCT116 was labeled with a lipophilic dye (DiI) and injected into the periviteline space (PVS) of 48 hpf zebrafish embryos (Figure 1A). The day after injection (1 dpi), xenotransplanted zebrafish larvae were either treated with 100 μg/mL deflamin (the highest tolerable dose) or left untreated. Deflamin was extracted, isolated, purified, and concentrated from dry Lupinus albus seeds as previously described [20]. E3 medium, containing or not containing deflamin was refreshed every day for a total of 3 days of treatment. At 4 dpi, animals were fixed and stained, and analyses of tumor size, apoptosis (activated caspase3), and proliferation (mitotic figures) were performed by confocal microscopy (Figure 1A). Results showed that deflamin treated tumors were on average four times smaller than untreated tumors (Figure 1B,C, *** *p* < 0.001). Importantly, analysis of cell death suggests that deflamin induces approximately a four-fold increase in apoptosis when compared to untreated tumors (Figure 1B,D, **** *p* < 0.0001). Tumor proliferation was not significantly affected by deflamin treatment (Figure 1B,E).

Moreover, this zebrafish xenograft model provides the opportunity to analyze metastasis formation given that at 4 dpi human fluorescently labeled tumor cells can be found in distant sites such as the brain, optic cup, gills, and caudal hematopoietic tissue (CHT). To assess the value of deflamin in metastasis formation, the number of xenografted zebrafish with micrometastasis was assessed. Results revealed that under deflamin treatment, HCT116 tumors had a reduced capacity to colonize secondary tissues (about 40% reduction in micrometastasis formation, Figure 1F,G).

Taken together, these results indicate that deflamin has an anti-tumor and anti-metastatic role in CRC development, suggesting important applications for cancer therapy.

### 3.2. Deflamin Does Not Play a Direct Role in Cancer Cell Proliferation or Apoptosis

To further investigate the mechanism of action of deflamin, we explored the direct effect of this molecule on cancer cell proliferation and apoptosis in vitro. To accomplish this, we treated HCT116, HT-29, and SW480 CRC cell lines with increased concentrations of deflamin (25 μg/mL, 50 μg/mL, and 75 μg/mL) for 5 days. The results showed no effect of this oligomeric protein on cancer cell proliferation rates, suggesting that deflamin does not play a direct role in cell cycle regulation (Figure 2A). Moreover, deflamin did not induce apoptosis of the HCT116, HT-29, and SW480 cells when caspase 3/7 activity was measured in vitro (Figure 2B), indicating that it does not hold a direct cytotoxic effect on cancer cells either. To further validate the absence of toxicity of deflamin in live organisms, we treated 72 hpf zebrafish larvae for three days with increased concentrations of deflamin (50 μg/mL and 100 μg/mL). Appendix A shows no mortality associated with zebrafish treatments under the conditions tested.

Overall, deflamin appears as a safe polypeptide oligomer to be administered in vivo. Furthermore, these results suggest that the increased apoptosis seen in the zebrafish model, was not caused by a direct effect of deflamin on cancer cells, but rather an indirect role, possibly through MMPs inhibition.

### 3.3. Deflamin Inhibits MMP-2 and MMP-9, Contributing to Impaired Cancer Cell Migration and Invasion

Although deflamin was not found to have a direct effect on cancer cell viability, gelatinases MMP-2 and MMP-9 are known to be critical for the ability of cancer cells to migrate and invade since they act not only on the degradation of the ECM (contributing to the rearrangement of the matrix and release of growth factors) but also on the degradation of cell–cell and cell–matrix adhesion molecules [3]. Zymographic analysis of MMPs from HCT116, HT-29, and SW480 CRC cell lines showed that deflamin exerts an inhibitory role on the activity of both MMP-2 and MMP-9 in all cell lines tested, displaying the greatest effect on SW480 cell line (a reduction of about 75% of MMPs activity at a deflamin concentration of 80 μg/mL, *** *p* < 0.001, Figure 2C). Therefore, we investigated the effect of deflamin on cancer cell migration using the wound healing in vitro assay (Figure 3A). Migration of cancer cells was shown to be impaired by the addition of deflamin to all cell lines tested, with the SW480 cell line showing the highest inhibition of cellular migration (about 77% reduction in migration at 80 μg/mL of deflamin concentration upon 72 h of treatment, ** *p* < 0.01, Figure 3A). We further tested the role of deflamin on cancer cell invasion using a 3D matrix of collagen (Figure 3B). MMP-2 and MMP-9 degrade collagen types III and I, respectively, being both able to cleave collagen types IV and V [28]. Therefore, analysis of 3D spheroids of the same cell lines showed a dose-dependent inhibition of invasion of 3D spheroids on a collagen matrix, being once more SW480 the cell line with the highest inhibition of invasive cells (reduction of about 40% invasiveness after 72 h of exposure to deflamin, *** *p* < 0.001, Figure 3B).

Overall, these results indicate that deflamin exerts an inhibitory effect on the activity of both MMP-2 and -9, as well as on cancer cell migration and invasion, with important implications for cancer development and progression.

### 3.4. Deflamin Inhibits Collagen Degradation and Angiogenesis In Vivo

For a tumor to continue to grow and start migrating/invading, two processes need to occur: (1) elimination of the physical barriers by ECM degradation; (2) generation of pro-angiogenic factors to allow the formation of new blood vessels. MMP-2 and MMP-9 are particularly important for both these processes since they increase the bioavailability of important factors such as vascular endothelial growth factor (VEGF), basic fibroblast growth factor (bFGF) and transforming growth factor β (TGF-β) by degrading ECM components such as collagen type IV and perlecan [29]. Thus, we investigated the role of deflamin in both of these processes through analysis of the extracellular collagen degradation and tumor angiogenesis in the zebrafish xenotransplant model. For that, HCT116 cells were stably transduced with a fluorescent dtTomato expressing vector and xenotransplanted into the PVS of wild type AB and Tg(kdrl:EGFP) zebrafish larvae (Figure 4A). For the analysis of collagen degradation in vivo, we used the collagen hybridizing peptide (CHP) which is a 5-FAM conjugated synthetic peptide that specifically binds to denatured collagen strands through hydrogen bonding (Figure 4B). CHP is an extremely specific probe for unfolded collagen molecules, while showing negligible affinity for intact collagen molecules due to a lack of binding sites [30]. In order to investigate blood vessel formation in the tumor area, the zebrafish Tg(kdrl: EGFP) model, which has the vessels labeled with GFP, was used [31] (Figure 4C). Using these models, our results show an inhibitory role of deflamin on collagen degradation when compared to control tumors in vivo (about 85% reduction in collagen degraded area, * *p* < 0.05, Figure 4D). Furthermore, analysis of tumor blood vessel formation showed about an 80% reduction in infiltrating vessels per tumor area (*** *p* < 0.001, Figure 4F).

Overall, our results indicate that, by inhibiting MMP-2 and -9 functions, deflamin impairs ECM remodeling through inhibition of collagen degradation and tumor angiogenesis, important processes for cancer progression, rather than having a direct action on cancer cell proliferation and apoptosis.

## 4. Discussion

Over the past decade, cancer pharmaceutics has been facing the challenge of maximizing the effectiveness and specificity of treatments, as well as minimizing the toxicity and resistance of therapeutic regimens. The increase in MMPs activity detected in a wide range of cancers has been taken as evidence for their implication in the cancer invasive and metastatic potential, therefore marking MMPs as important targets for both diagnostic and therapeutic purposes [5]. This feature has been well demonstrated in several works, comprising selective inhibition and MMP-9-deficient mice, all of which pointed to MMP-9 as an important target of neoplastic diseases [32]. Indeed, in recent years a substantial amount of research has been made, attempting to develop synthetic, low-molecular-weight inhibitors of MMPs (MMPIs) for the potential treatment of diseases in which they play a major role. However, technical difficulties, side effects, and dose-dependent toxicity have greatly limited the success of these anti-MMP drugs [8,9,10,11]. Nevertheless, interesting results have been obtained with natural compounds with anti-inflammatory and anti-tumoral activity. Currently, studies on molecules of natural origin have shown promise in inhibiting MMPs, especially MMP-9, in inflammatory and oncogenic pathological processes [12,13,14,15,16,17]. Deflamin is a natural food component extracted from white lupine seeds (*Lupinus albus*) that shows anti-MMP-2 and MMP-9, as well as anti-inflammatory activities [20,21,22,23]. Importantly, deflamin has the advantage of being a water-soluble molecule easily extracted and isolated in vitro, that shows resistance to boiling and to digestive enzymatic reactions and has the potential to act locally in the intestinal system (without being absorbed into circulation), likely bypassing the problem of systemic toxicity associated with common MMPIs [20,21,22,23]. In this sense, this work aimed to explore deflamin therapeutic potential, using 3D cellular systems and zebrafish larvae models. Even though drug pharmacodynamics in zebrafish may differ from mammals, many compounds have been shown to block disease in a similar way in both organisms [33]. Therefore, this work corroborated that deflamin does not hold a cytotoxic effect on the CRC cell lines and zebrafish embryos, providing evidence for its safety as a potential therapeutic strategy. Moreover, deflamin showed inhibitory activity of the invasive process both in cellular systems and in zebrafish cancer models. Importantly, deflamin was effective in inhibiting MMP-2, MMP-9, and general ECM remodeling, favoring spatial constraints of tumor growth/progression and limited nutrient/oxygen supply, due to decreased angiogenesis. In agreement with this work, our previous studies have suggested that deflamin’s mode of action involves direct inhibition of MMP-9 and -2, due to its biochemical features [21]. Hence, rather than inhibiting any of the regular pathways of MMP activation or expression, deflamin reduces gelatinase activity, without inducing direct alterations in the cell cycle or in gene expression. However, by limiting ECM remodeling, deflamin prevents the angiogenic process, as well as the degradation of the physical barriers necessary for tumor growth. In this context, physical constraint renders tumors unable to proliferate and metastasize, finishing by becoming apoptotic. Thus, by reducing gelatinase activities in situ, deflamin provides a simple manner to restrict the tumor and render it unable to progress.

Although the present work has some limitations as the zebrafish model is limited in terms of its similarity to mammal models, our results, paired with our previous findings on deflamin, bring a novel view on the use of MMP-2 and -9 inhibitors in cancer models. Indeed, despite MMP-9 being known for decades as an attractive target for anticancer therapies, the development of effective and safe MMP9 inhibitors as anticancer drugs has been shown to be extremely difficult. Recently, therapies have been aiming at more specificity through blocking antibodies that selectively inactivate MMP9, and these are currently in clinical trials [34]. It seems however that the feature of being able to reduce gelatinolytic activity directly and in situ by a food component such as deflamin might be a similar and simpler approach for tackling cancer disease via MMP inhibition.

Overall, although MMP9 has been well established as an important target in anticancer treatments, there is still a need for selective, safe, and effective MMP-9 inhibitors. This is the first report on an effective food protein gelatinase inhibitor that reduces cancer development via a constriction of the tumor’s 3D space distribution. Added to the fact that it is of food origin and easy to isolate, this work revealed the nutritional potential of deflamin as a co-adjuvant therapeutic agent in the treatment of CRC, as a nontoxic dietary supplement. Further work using more complex animal models of cancer and pre-clinical trials will, with no doubt, bring novel insights into the effectiveness of deflamin against CRC.

## 5. Conclusions

The work presented here demonstrated that deflamin was able to impair CRC angiogenesis and tumor microenvironment remodeling, via gelatinase inhibition, which led to a constriction and limitation of the tumor’s spatial distribution and nutrient and oxygen supply. This type of mechanism seems to be a good depiction of what a specific MMP inhibitor should attain by reducing gelatinase activity, angiogenesis is efficiently impaired limiting tumor growth and inducing cancer cell apoptosis.

Our results suggest that being a natural compound, non-toxic, and resistant to digestion, deflamin holds the potential to be a novel nutraceutical, adjuvant or even a functional food to be used in treating and preventing CRC via a specific, in situ, gelatinase inhibition. Further studies of the use of this oligomeric protein are needed to assess its clinical and commercial value, but it seems plausible to infer that using this type of gelatinase inhibition could be a novel and effective approach to tackling gastrointestinal cancer disease.

## Figures and Tables

**Figure 1 cancers-14-06182-f001:**
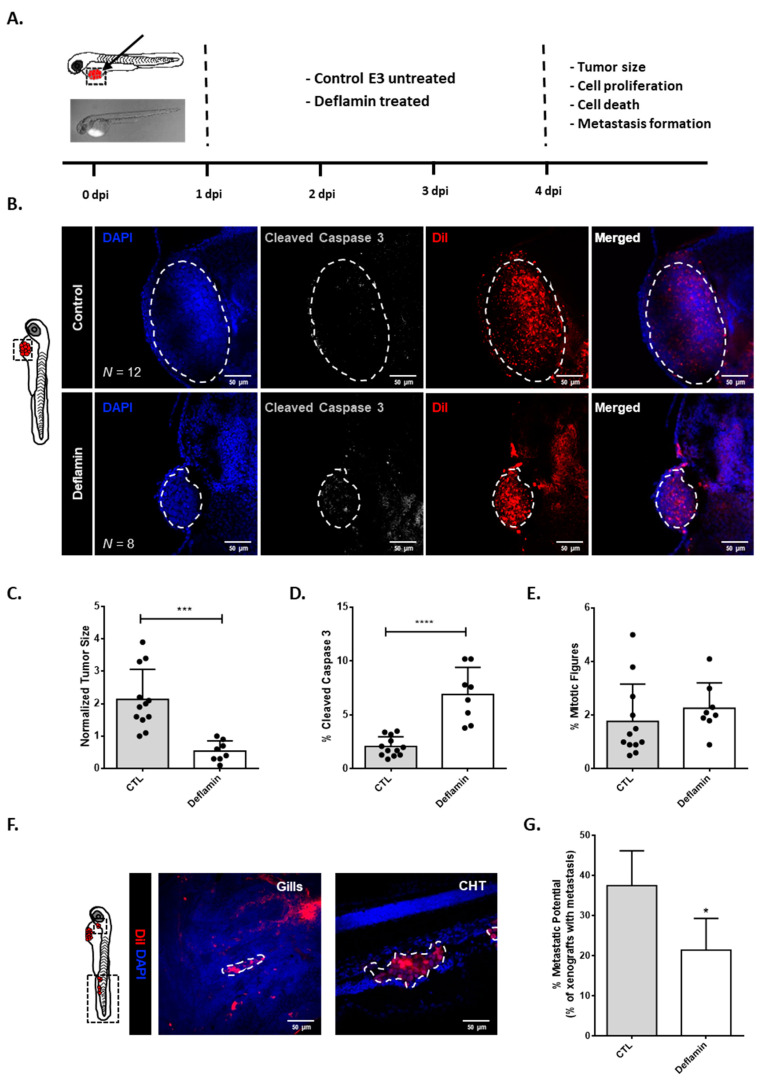
Zebrafish xenotransplant model of HCT116 cells exposed to deflamin: (**A**) Human cancer cell line HCT116 was fluorescently labeled with DiI (red) and injected into the perivitelline space (PVS) 2 days post-fertilization (dpf) nacre/casper zebrafish larvae. Zebrafish xenografts were treated in vivo with deflamin for 72 h and compared with untreated controls regarding tumor size, cell death, cell proliferation, and metastasis formation; (**B**) At 4 days post-injection (dpi), zebrafish xenografts were imaged on PVS by confocal microscopy; (**C**) Analysis of tumor size (***, *p* ≤ 0.001); (**D**) Analysis of activated caspase 3 (apoptosis, ****, *p* ≤ 0.0001); (**E**) Analysis of mitotic figures (proliferative cells, ns); (**F**) Zebrafish xenografts were also imaged over the entire body by confocal microscopy. Representative images of HCT116 micrometastasis in gills and caudal hematopoietic tissue (CHT); (**G**) % of zebrafish exhibiting metastasis. The number of xenografts analyzed is indicated in the representative images. In the graphs, each dot represents one zebrafish xenograft. Statistical analysis was performed as described in the Statistical Analysis section (*, *p* ≤ 0.05, ***, *p* ≤ 0.001, ****, *p* ≤ 0.0001). Scale bars represent 50 μm. All images are anterior to the left, posterior to the right, dorsal up, and ventral down.

**Figure 2 cancers-14-06182-f002:**
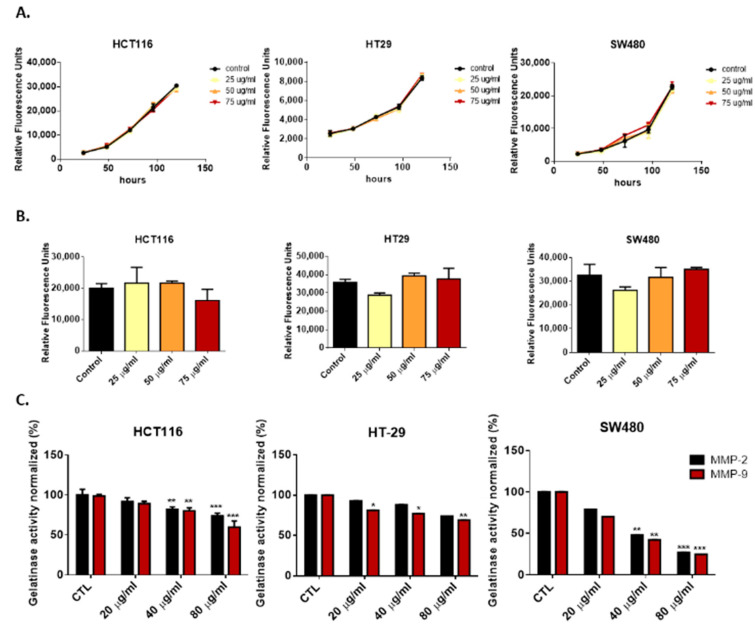
Effect of Deflamin on viability and MMPs of CRC cell lines HCT116, HT-29, and SW480: (**A**) Viability of human CRC cell lines when exposed to deflamin; (**B**) Analysis of cell death by apoptosis of human CRC cell lines treated with deflamin; (**C**) Zymographic analysis of the anti-gelatinases activity of deflamin. Statistical analysis was performed as described in the Statistical Analysis section (*, *p* ≤ 0.01; **, *p* ≤ 0.01, ***, *p* ≤ 0.001).

**Figure 3 cancers-14-06182-f003:**
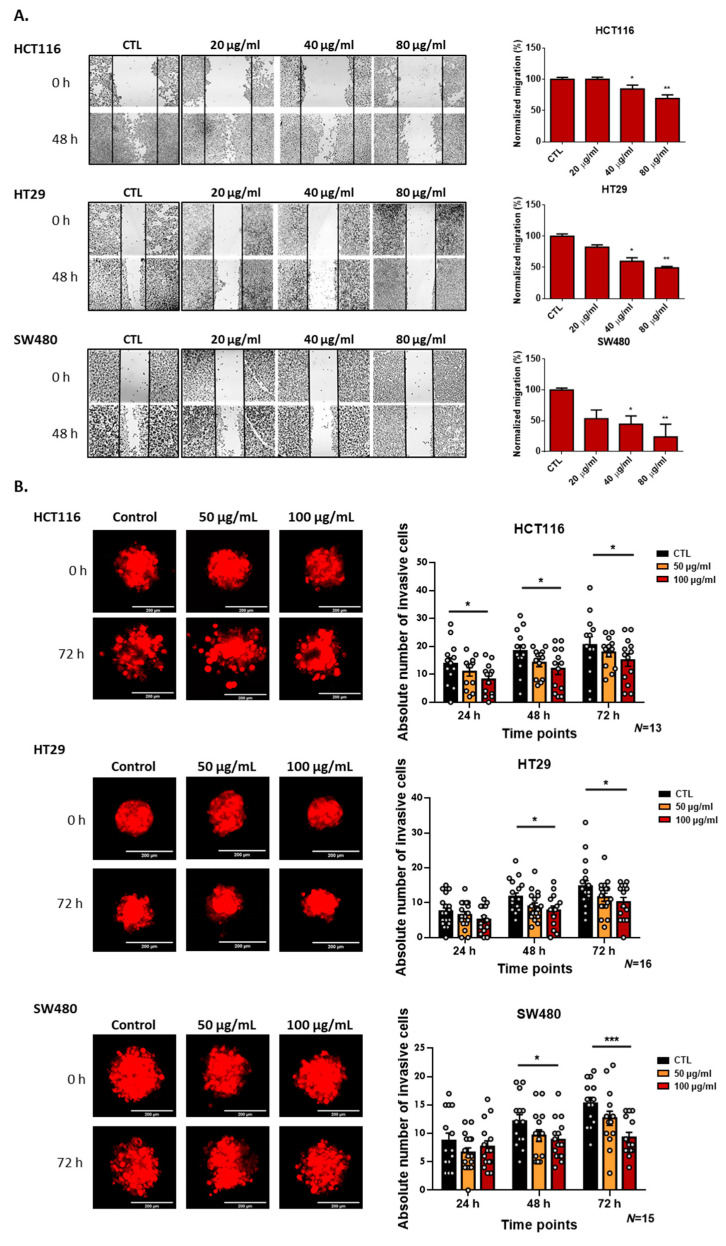
Migration and invasion of CRC cell lines HCT116, HT-29, and SW480 upon treatment with deflamin: (**A**) Analysis of migration by the wound healing assay of human CRC cell lines HCT116, HT-29 and SW480 treated with deflamin for 48 h and its corresponding quantification (%) (*n* = 3). (**B**) 3D spheroid invasion assay of human CRC cell lines HCT116, HT-29, and SW480 treated with deflamin and the corresponding quantification (number of absolute invasive cells). The number of spheroids analyzed is indicated in the graphs. Statistical analysis was performed as described in the Statistical Analysis section (*, *p* ≤ 0.05; **, *p* ≤ 0.01; ***, *p* ≤ 0.001).

**Figure 4 cancers-14-06182-f004:**
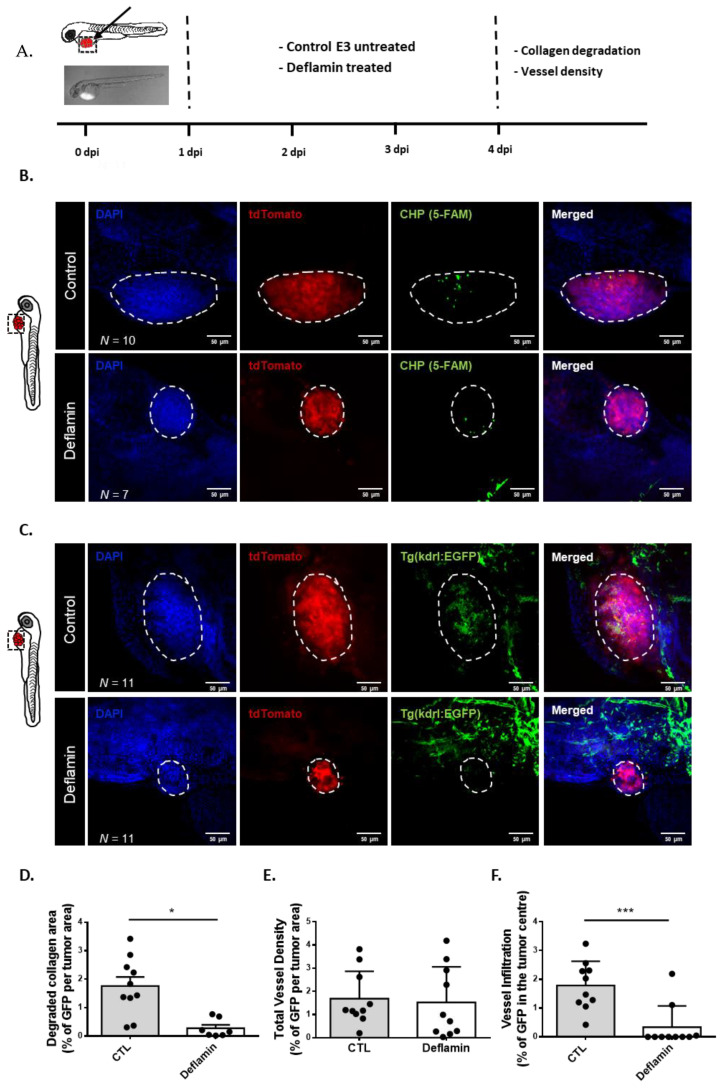
Cancer microenvironment analysis of HCT116 zebrafish xenotransplants: (**A**) Human cancer cell line HCT116 was stably transduced with FUdtTW plasmid (tdTomato expression marker) and injected into the perivitelline space (PVS) of 2 days post-fertilization (dpf) nacre/casper zebrafish larvae. Zebrafish xenografts were treated in vivo with deflamin for 72 h and compared with untreated controls regarding collagen degradation and vessel density; (**B**,**C**) At 4 days post-injection (dpi), zebrafish xenografts were imaged on PVS by confocal microscopy; (**D**) Quantification of degraded collagen area by analysis of CHP staining (5-FAM stained area); (**E**) Total vessel density analysis by EGFP marker; (**F**) Tumor vessel infiltration analysis by EGFP marker. The number of xenografts analyzed is indicated in the representative images. In the graphs, each dot represents one zebrafish xenograft. Statistical analysis was performed as described in the Statistical Analysis section (*, *p* ≤ 0.05; ***, *p* ≤ 0.001). Scale bars represent 50 μm. All images are anterior to the left, posterior to the right, dorsal up, and ventral down.

## Data Availability

The data presented in this study are available on request from the corresponding author.

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
