# Peer review of "Therapeutic Potential of Deflamin against Colorectal Cancer Development and Progression"

_cancers, 2022, doi:10.3390/cancers14246182_

Round 1
Reviewer 1 Report
The manuscript was prepared very well. The introduction section justifies the purpose of the study. I congratulate the authors for the preparation of the manuscript
I would like to congratulate the authors for the structure of the manuscript and all the research carried out. It is highly publishable. However, there are some concerns, in part important, so the review articles need revision, see below.
Introduction
Explain something more about deflamin. Has it been used as a nutraceutical in any other disease?
Why do you think it has potential as an anti-cancer drug?
Would it be used as an adjuvant treatment to standard therapies?
Materials and Methods
The methodology is perfectly described and carried out
Have you registered it on a platform like ClinicalTrials.gov?
Results
· The tables and the text describing them do not require any input, it is the strongest part of this study.
Discussion
· what does this manuscript specifically contribute?
· Include a limitations /strengths section.
· Although there is an excellent description of the studies and description of the results, possible mechanisms related to the results described in the manuscript could be added.
· In the Conclusion section, state the most important outcome of your work. Do not simply summarize the points already made in the body — instead, interpret your findings at a higher level of abstraction. Show whether, or to what extent, you have succeeded in addressing the need stated in the Introduction (or objectives).
Reviewer 2 Report
This study by Silva et al. demonstrated that deflamin, a plant protein isolated from the white lupine seeds, exerted anti-tumor effect in a zebrafish model of tumorigenesis and metastasis. Mechanistically, deflamin inhibits MMP activity cancer cell lines in vitro and impairs cancer cell migration and invasion. Overall, this study is built on this group’s previous work and describes a potential critical translational application of deflamin, however, several major points should be addressed.
The following major points are raised:
1. In Figure 1B, the IF staining of cleaved caspase 3 was very weak thus challenging to analyze or interpretate. Though the authors draw conclusion that deflamin administration increased apoptosis of cancer cells in vivo. This piece of data should be clarified/improved before reaching any conclusions on deflamin and cancer cell apoptosis.
2. Using in vitro culture system, the authors showed that deflamin treatment did not impact proliferation and apoptosis of cancer cell lines (Figure 2). However, this is in contradictory with the in vivo data, please further clarify or include this in the discussion.
3. The authors conclude that deflamin appears to be safe to be administered in vivo based on the in vitro data in Figure 2, however, one could argue that cancer cell lines or primary cancer cells behave and metabolize differently from normal tissue cells, the authors should at least temper this claim regarding to possible side effects of deflamin administration in vivo.
3. The authors utilized a 3D collagen matrix to demonstrate the regulation of MMP2/9 activity by deflamin, however, the representative images showed minimal differences between control and deflamin-treated groups (Figure 3B), and the statistics also showed a very minor but significant effect. This is unusual given that the in vivo phenotype in Figure 1 is much stronger. Is it possible that this assay is not sensitive enough or could there be other factors that work cooperatively with deflamin for anti-tumor effect in vivo?
4. Lastly, the authors demonstrated that deflamin administration in vivo resulted in less degraded collagen in the implanted tumor, however, the CHP(5-FAM) IF staining is not convincing.
5. In general, the link between deflamin regulation of MMP activity and anti-tumor effect is weak. Is it possible to use a MMP KO model to show a causal relationship?
Minor concerns:
1. Figure 1F, please show representative IF images for control and deflamin treated groups. Also, statistical analysis is missing in the bar graph.
2. In Figure 4C, there is one circle missing in one of the images.
Round 2
Reviewer 2 Report
Thanks the authors for their efforts in attempting to address the points that reviewers raised. However, it is concerning that the edited manuscript has a weird format that Figure 2 is missing and Figure 3 is duplicated, which requires further editing.
To Reviewer 2 point 5, the authors still did not address the causal relationship between deflamin's regulation of MMP activity in vitro and deflamin's anti-tumor effect in zebrafish in vivo. The authors should either provide more evidence such as using a MMP inhibitor/MMP KO animal model to demonstrate the causal relationship or temper their claim in the summary (lines 18-19).
Round 3
Reviewer 2 Report
The authors have addressed the points raised in the last round of review.